# Long Range Arena: A Benchmark for Efficient Transformers

**Yi Tay**[1*]**, Mostafa Dehghani**[1*]**, Samira Abnar**[1]**, Yikang Shen**[1]**, Dara Bahri**[1]**, Philip Pham**[1]
**Jinfeng Rao**[1]**, Liu Yang**[1]**, Sebastian Ruder**[2]**, Donald Metzler**[1]
[1]Google Research
[2]Google DeepMind
{yitay, dehghani}@google.com

## Abstract

Transformers do not scale very well to long sequence lengths largely because of quadratic self-attention complexity. In the recent months, a wide spectrum of efficient, fast Transformers have been proposed to tackle this problem, more often than not claiming superior or comparable model quality to vanilla Transformer models. To this date, there is no well-established consensus on how to evaluate this class of models. Moreover, inconsistent benchmarking on a wide spectrum of tasks and datasets makes it difficult to assess relative model quality amongst many models. This paper proposes a systematic and unified benchmark, *Long-Range Arena*, specifically focused on evaluating model quality under long-context scenarios. Our benchmark is a suite of tasks consisting of sequences ranging from $1K$ to $16K$ tokens, encompassing a wide range of data types and modalities such as text, natural, synthetic images, and mathematical expressions requiring similarity, structural, and visual-spatial reasoning. We systematically evaluate ten well-established long-range Transformer models (Reformers, Linformers, Linear Transformers, Sinkhorn Transformers, Performers, Synthesizers, Sparse Transformers, and Longformers) on our newly proposed benchmark suite. Long-Range Arena paves the way towards better understanding this class of efficient Transformer models, facilitates more research in this direction, and presents new challenging tasks to tackle.

## 1 Introduction

Transformers (Vaswani et al., 2017) are ubiquitously state-of-the-art across many modalities, from language (Devlin et al., 2018; Raffel et al., 2019; Child et al., 2019) to images (Tan & Bansal, 2019; Lu et al., 2019) to protein sequences (Rives et al., 2019). A common weakness of Transformers is their quadratic memory complexity within the self-attention mechanism that restricts their potential application to domains requiring longer sequence lengths. To date, a dizzying number of efficient Transformer models ('*xformers*') have been proposed to tackle this problem (Liu et al., 2018; Kitaev et al., 2020; Wang et al., 2020; Tay et al., 2020b; Katharopoulos et al., 2020). Many of these models demonstrate comparable performance to the vanilla Transformer model while successfully reducing the memory complexity of the self-attention mechanism. An overview of this research area can be found in (Tay et al., 2020c).

Comparing the evaluation and experimental setup of many of these papers, we can make the following observations. Firstly, there is no unifying consensus on what makes an acceptable test bed for benchmarking efficient Transformers. There is also a large diversity in the types of tasks adopted—every single model is evaluated on a different set of tasks and datasets, which makes comparison of different models as well as an assessment of their relative strengths and weaknesses difficult. Secondly, the benchmarks used for evaluation are often arbitrarily chosen, without much consideration to whether the task is suitable for evaluating long-range modeling. Thirdly, many papers tend to conflate the effectiveness of the inductive bias with the benefits of pretraining (Ainslie et al., 2020; Zaheer et al., 2020; Wang et al., 2020), which tends to obfuscate the true value of the architecture.

---

[*]First two authors contributed equally.

Pretraining itself is a computationally expensive endeavour and de-coupling inductive bias research from pretraining would make xformer research more accessible.

In this paper, we propose a new benchmark, *Long-Range Arena* (LRA), for the purpose of benchmarking sequence models under the long-context scenario. We design a benchmark suite comprised of both synthetic probing tasks and real-world tasks and provide relative comparisons for **ten** recently proposed efficient Transformer models including Sparse Transformers (Child et al., 2019), Reformer (Kitaev et al., 2020), Linformer (Wang et al., 2020), Longformer (Beltagy et al., 2020), Sinkhorn Transformers (Tay et al., 2020b), Performers (Choromanski et al., 2020), Synthesizers (Tay et al., 2020a), Linear Transformers (Katharopoulos et al., 2020), and BigBird (Zaheer et al., 2020). This is the most comprehensive and extensive side-by-side evaluation of this class of models.

While the focus of this benchmark is the ability of these architectures to reason in long-context scenarios, we are also fundamentally interested in understanding the capabilities and properties of these xformer architectures when exposed to different types of data and conditions. Hence, our benchmark is purposefully designed to be capability probing, i.e, we select datasets and tasks with certain innate structure. For example, can these architectures model long sequences that are intrinsically hierarchical or that contain some form of spatial structure? In general, we are especially interested in the relative performance of these xformer models across diverse circumstances. We hope that understanding these better will inspire research on more efficient architectures in the future. While the focus of this paper is on efficient Transformer models, our benchmark is also model agnostic and can also serve as a benchmark for long-range sequence modeling.

Aside from comparing the quality of these models, we also conduct extensive efficiency and memory usage analysis of these models. We believe such a side-by-side performance benchmark will be valuable to the community, providing deeper insight on the practical efficiency of these methods.

Overall, we propose a unified framework for enabling easy side-by-side comparisons of efficient Transformer models and broadly speaking, long-range sequence models in general. Our framework, which we plan to open source, is written in JAX/FLAX[1].

## 2 LONG-RANGE ARENA (LRA)

This section introduces the Long-Range Arena (LRA) benchmark (pronounced *el-ra*). We implement our benchmark (which includes the task, evaluators, and models) in Python 3 and Jax/Flax and plan to open-source our code—making it easy to extend and to build on top of our work.

### 2.1 DESIDERATA

For creating the Long-Range Arena benchmark, we established a set of desiderata:

1. **Generality**: All efficient Transformers models should be applicable to our tasks. For instance, given that not all xformer models are able to perform autoregressive decoding (Wang et al., 2020), we include tasks that only require encoding.

2. **Simplicity**: The tasks should have a simple setup. All factors that make comparisons difficult should be removed. This encourages simple models instead of cumbersome pipelined approaches. For instance, we avoid including any particular data augmentation and consider pretraining to be out of scope of this benchmark.

3. **Challenging**: The tasks should be difficult enough for current models to ensure there is room for improvement to encourage future research in this direction.

4. **Long inputs**: The input sequence lengths should be reasonably long since assessing how different models capture long-range dependencies is a core focus of LRA.

5. **Probing diverse aspects**: The set of tasks should assess different capabilities of models like their ability to model relations and hierarchical/spatial structures, generalization capability, etc.

6. **Non-resource intensive and accessible**: The benchmarks should be deliberately designed to be lightweight so as to be accessible to researchers without industry-grade computing resources.

---

[1]https://github.com/google/flax

## 2.2 TASKS

This section describes the tasks in the LRA benchmark. Note that these tasks are specifically designed for the purpose of assessing different aspects of efficient Transformer models. Further details about each task can be found in the appendix.

### 2.2.1 LONG LISTOPS

In this task, we are interested in the capability of modeling hierarchically structured data in a long-context scenario. This benchmark task is a longer variation of the standard ListOps task proposed in (Nangia & Bowman, 2018), which was designed to investigate the parsing ability of neural models.

The dataset is comprised of sequences with a hierarchical structure and operators MAX, MEAN, MEDIAN and SUM_MOD that are enclosed by delimiters (brackets). An example (much shorter) sequence is as follows:

**INPUT:** [MAX 4 3 [MIN 2 3 ] 1 0 [MEDIAN 1 5 8 9, 2]]     **OUTPUT:** 5

In our task we use a version of ListOps of sequence lengths of up to $2K$ to test the ability to reason hierarchically while handling long contexts. Naturally, in the above example the model needs to access all tokens and model the logical structure of the inputs in order to make a prediction. The task is a ten-way classification task and is considerably challenging.

### 2.2.2 BYTE-LEVEL TEXT CLASSIFICATION

This task using real-world data represents a common use case of efficient Transformers, which are often needed to process long documents. Text classification in particular is associated with many real-world applications such as spam, fraud, and bot detection and commercial document classification, among others (Howard & Ruder, 2018).

This task also benchmarks the ability of the models to deal with compositionality as it is required to compose characters into words into higher-level phrases. Compared to ListOps, boundaries are less well defined and need to be learned from the data, which is a challenging problem in its own right (Kawakami et al., 2019).

We consider the byte/character-level setup of this task in order to simulate a longer input sequence, which also makes the task considerably more challenging.[2] Note that this setup differs significantly from character-level language modeling (char LM). In char LM, it would suffice to read nearby context to determine the next character, e.g., a model is very likely to predict *'e'* after having seen the prefix *'appl'*. For byte-level text classification, the model needs to reason with compositional, unsegmented data in order to solve a meaningful real-world task. We use the IMDb reviews (Maas et al., 2011) dataset, which is a commonly used dataset to benchmark document classification. We use a fixed max length of $4K$ for this task, which is truncated or padded when necessary. This is a binary classification task with accuracy as the metric.

### 2.2.3 BYTE-LEVEL DOCUMENT RETRIEVAL

We further evaluate a model's ability to encode and store compressed representations that are useful for matching and retrieval. Learning the similarity score between two vectors is a common problem in machine learning and is useful for a wide array of applications (Guo et al., 2016). Hence, this task is mainly about modeling a similarity score between two documents in a 'two tower setup' in which compressed representations are concatenated and passed into a linear classifier. Note that we deliberately prevent models from using cross attention. This task thus serves as a test of how well models are able to *compress* long sequences into representations suitable for similarity-based matching.

We use the ACL Anthology Network (AAN; Radev et al., 2013) dataset, which identifies if two papers have a citation link, a common setup used in long-form document matching (Jiang et al.,

---

[2]On the IMDb word-level task, models without pre-training achieve accuracies in the high 80s while the same models score in the mid 60s on the character-level task (Tay et al., 2020b).

2019; Yang et al., 2020). Similar to the text classification setup, we use a byte/character level setup, which challenges the model to compose and aggregate information over longer contexts. We use a sequence length of $4K$ for each document, which makes the total text length $8K$ for this task. This is a binary classification task with accuracy as the metric.

### 2.2.4 IMAGE CLASSIFICATION ON SEQUENCES OF PIXELS

This task is an image classification task, where the inputs are sequences of pixels. In other words, an $N \times N$ image is flattened to a sequence of length $N^2$ pixels. Similar to how the previous tasks require capturing the hierarchical structure in the data, this task requires the model to learn the 2D spatial relations between input pixels, while presented as a 1D sequence of symbols. We focus on assessing Transformer models that are designed to process a sequence of discrete symbols, so we do not allow extra modules such as a CNN stem that embeds pixel-level inputs. To simplify the setup, we map the input images to a single gray-scale channel where each pixel is represented with an 8-bit pixel intensity (vocabulary size of 256). In LRA, we use the CIFAR-10 dataset (Krizhevsky, 2009) for the image classification task.

### 2.2.5 PATHFINDER (LONG-RANGE SPATIAL DEPENDENCY)

The Pathfinder challenge (Linsley et al., 2018; Kim* et al., 2020) was first introduced for learning long-range spatial dependencies. It is a synthetic visual task motivated by cognitive psychology (Houtkamp & Roelfsema, 2010). The task requires a model to make a binary decision whether two points represented as circles are connected by a path consisting of dashes. We show a positive example of two connected points and a negative example of two unconnected points in Figure 1.

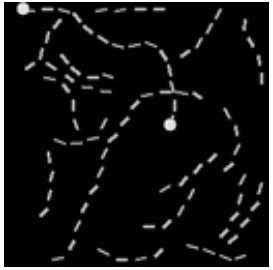

(a) A positive example.

The dataset also contains distractor paths, which makes this setup challenging. We model this task by treating images as sequences of pixels. In this task, images are of dimensions ($32 \times 32$), which make up a sequence length of $1024$.

### 2.2.6 PATHFINDER-X (LONG-RANGE SPATIAL DEPENDENCIES WITH EXTREME LENGTHS)

Finally, we consider an extreme version of Pathfinder (Pathfinder-X) where examples consist of $16K$ pixels (i.e., images of $128 \times 128$). The key goal here is to observe if a model would fail to solve the $16K$ extreme version even if it can successfully learn the standard version of $1024$ tokens. This is an interesting litmus test to see if the same algorithmic challenges bear a different extent of difficulty when sequence lengths are much longer. We include this in our benchmark as Path-X.

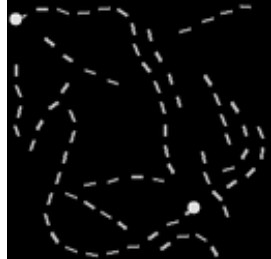

(b) A negative example.

Figure 1: Samples of the Pathfinder task.

### 2.3 REQUIRED ATTENTION SPAN OF LRA TASKS

One of the main goals of the LRA benchmark is assessing the ability of different efficient Transformer models to capture long-range dependencies. The tasks and setups are designed with this goal in mind. In order to have a quantitative estimate of the spatial extent needed to be considered by an attention mechanism to encode the inputs, we define *required attention span*.

Given a trained attention-based model and a sequence of tokens as inputs, the required attention span of an attention module is computed as the mean distance between the query token and the attended tokens, scaled by attention weights. Here, we compute the mean *required atten-*

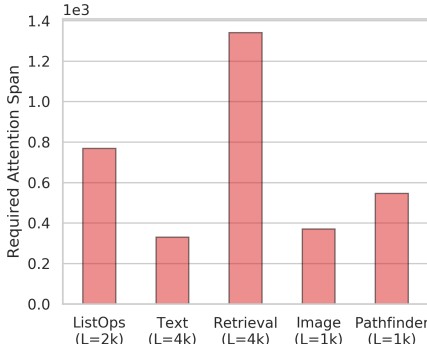

Figure 2: Required attention span on different tasks.

| Model | ListOps | Text | Retrieval | Image | Pathfinder | Path-X | Avg |
|---|---|---|---|---|---|---|---|
| Chance | 10.00 | 50.00 | 50.00 | 10.00 | 50.00 | 50.00 | 44.00 |
| Transformer | 36.37 | 64.27 | 57.46 | 42.44 | 71.40 | FAIL | 54.39 |
| Local Attention | 15.82 | 52.98 | 53.39 | 41.46 | 66.63 | FAIL | 46.06 |
| Sparse Trans. | 17.07 | 63.58 | **59.59** | **44.24** | 71.71 | FAIL | 51.24 |
| Longformer | 35.63 | 62.85 | 56.89 | 42.22 | 69.71 | FAIL | 53.46 |
| Linformer | 35.70 | 53.94 | 52.27 | 38.56 | 76.34 | FAIL | 51.36 |
| Reformer | **37.27** | 56.10 | 53.40 | 38.07 | 68.50 | FAIL | 50.67 |
| Sinkhorn Trans. | 33.67 | 61.20 | 53.83 | 41.23 | 67.45 | FAIL | 51.39 |
| Synthesizer | 36.99 | 61.68 | 54.67 | 41.61 | 69.45 | FAIL | 52.88 |
| BigBird | 36.05 | 64.02 | 59.29 | 40.83 | 74.87 | FAIL | **55.01** |
| Linear Trans. | 16.13 | **65.90** | 53.09 | 42.34 | 75.30 | FAIL | 50.55 |
| Performer | 18.01 | 65.40 | 53.82 | 42.77 | **77.05** | FAIL | 51.41 |
| Task Avg (Std) | 29 (9.7) | 61 (4.6) | 55 (2.6) | 41 (1.8) | 72 (3.7) | FAIL | 52 (2.4) |

Table 1: Experimental results on Long-Range Arena benchmark. Best model is in boldface and second best is underlined. All models do not learn anything on Path-X task, contrary to the Pathfinder task and this is denoted by FAIL. This shows that increasing the sequence length can cause seriously difficulties for model training. We leave Path-X on this benchmark for future challengers but do not include it on the Average score as it has no impact on relative performance. **Important: These results always represent the results at the time of ICLR submission and are not modified for archival purposes. Please see appendix in arxiv version of this paper for updated snapshots or new runs of this leaderboard with new settings and use that for paper comparisons.**

*tion span* over all attention modules in our best vanilla Transformer model for each task, averaged over 1K random samples from the validation set. Figure 2 summarizes the required attention span for each task in LRA. For all the tasks in LRA the required attention span is rather high. This shows, a Transformer model needs to go beyond combining only local information, while in many other tasks and datasets, attention mechanism mostly need to combine information from neighboring positions. Given the purpose of LRA, we found *required attention span* serves as a good proxy for how difficult a task is for Transformer-based models.[3]

## 3 EXPERIMENTAL RESULTS

### 3.1 MODELS

This section describes the models we evaluate on our LRA benchmark. We base our evaluation on ten recently proposed efficient Transformer models. Aside from the standard vanilla Transformer (Vaswani et al., 2017) and a simple local attention baseline, we compare Sparse Transformers (Child et al., 2019), Longformers (Beltagy et al., 2020), Linformers (Wang et al., 2020), Reformers (Kitaev et al., 2020), Sinkhorn Transformers (Tay et al., 2020b), Synthesizers (Tay et al., 2020a), BigBird (Zaheer et al., 2020), Linear Transformers (Katharopoulos et al., 2020), and Performers (Choromanski et al., 2020). We believe these ten models to represent a diverse cross-section of recent efficient Transformer models.

### 3.2 PHILOSOPHY BEHIND THE BENCHMARK

We note that it is non-trivial and almost impossible to conduct a perfectly fair evaluation of all models. The large search space motivates us to follow a set of fixed hyperparameters (number of layers, heads, embedding dimensions, etc.) for all models. The best performance and relative order of the models may change if we aggressively tune hyperparameters for all models. Hence, the results provided in this paper are not meant to be a final authoritative document on which xformer is the

---

[3]Note that this metric mainly provides an indication of the required attention span for a task and the relative differences between tasks based on a reasonably strong model; a better model might only need to attend to shorter ranges (Daniluk et al., 2017; Rae & Razavi, 2020).

best. Instead, we provide a starting point for future research and strive to be as **fair** as possible. In order to do so, we plan to release the code with all the hyperparameters and implementation details. Additionally, we intend for our paper to be a living document and encourage researchers (authors and the broader community) to contribute and continue updating this paper (with rules and limitations described in the appendix). We also implemented all models to the best of our abilities. We often consulted with the original developers of the included models. For simplicity, the main text of the paper will always contain the initial set of results that we submitted to ICLR. **Important: All future updates will, for consistency, be in the appendix of the paper with marked version identifiers so other researchers can always reference them in other papers. If you obtain a better result on any of the following models we tried, please send an email and we can discuss making an update to the leaderboard.**

### 3.3 QUANTITATIVE RESULTS

Based on our results, we observe that (1) all proposed tasks in LRA are considerably challenging and (2) there are meaningful differences in model performance across different xformer models.

**Results on ListOps**  The ListOps task (10-way classification) has proven to be reasonably difficult with the best models obtaining only $37\%$. The considerable gap to random chance shows that models are indeed learning the task. We notice that the inductive bias of the xformer models plays a substantial role on this task in which approximately half the xformer models are able to get $> 30\%$ performance while the remainder of the models only get slightly above random chance. This may imply that certain efficiency-inspired inductive biases may be better at handling hierarchical data than others. For instance, the results from our experiments seem to suggest that kernel-based models (e.g., Performer, Linear Transformers) are possibly not as effective on hierarchically structured data. We feel that ListOps may be useful in probing a model's capability in handling hierarchically structured data.

**Results on Text Classification**  Byte-level classification is shown to be difficult and challenging especially when no pretraining or contextual embeddings are used. The best model only obtains 65.90 accuracy. The Linear Transformer performs well on this task, along with the Performer model. Contrary to the ListOps task, it seems like fast kernel-based models do well on this task.

**Results on Retrieval**  The scores of different models on this task are also rather low (average of $55\%$), indicating the difficulty of the task. The vanilla Transformer model only achieves $57.46\%$ accuracy with some xformer variants scoring very close to random chance. The best performing model is the Sparse Transformer and the second best is BigBird. We find that models that follow fixed sparse patterns to do well on this task. Models that are based on low-rank factorization and kernels perform relatively worse.

**Results on Image Classification**  On the image classification task, most models perform quite similarly (low variance amongst model performance). The best model on this task is the Sparse Transformer, followed by the Performer. Linformer and Reformers do not do well on this task. On a related note, we also observed most of models struggle generalizing to the test even though they manage to overfit the training set. While we extensively tried different regularization techniques on every single model, there is a rather large gap between their performance on train and test set (More details in Appendix).

**Results on Pathfinder / Path-X**  Results show that all models achieve reasonable performance on the Pathfinder task. The average performance is 72 and the best model Performer obtains $77.05\%$ accuracy. The Local Attention model performs the worse out of all models. It seems that the best models on this spatial reasoning task are the kernel models (Performer and Linear Transformer).

All models failed to solve the Path-X task, achieving at best $50\%$. We find this intriguing because this is essentially an identical task to the standard Pathfinder, albeit with much longer sequence lengths. Hence, we observe that the extreme length of the task can significantly obstruct a model from leaning anything meaningful. We leave Path-X in our benchmark suite, hoping to spur future progress in modeling sequences at extreme lengths.

| Model | Train Speed (Steps per second) | | | | Peak Memory Usage (GB) | | | |
|---|---|---|---|---|---|---|---|---|
| | 1K | 2K | 3K | 4K | 1K | 2K | 3K | 4K |
| Transformer | 8.1 | 4.9 | 2.3 | 1.4 | 0.85 | 2.65 | 5.51 | 9.48 |
| Local Attention | 9.2 (1.1x) | 8.4 (1.7x) | 7.4 (3.2x) | 7.4 (5.3x) | 0.42 | 0.76 | 1.06 | 1.37 |
| Linformer | 9.3 (1.2x) | 9.1 (1.9x) | 8.5 (3.7x) | 7.7 (5.5x) | **0.37** | **0.55** | 0.99 | **0.99** |
| Reformer | 4.4 (0.5x) | 2.2 (0.4x) | 1.5 (0.7x) | 1.1 (0.8x) | 0.48 | 0.99 | 1.53 | 2.28 |
| Sinkhorn Trans | 9.1 (1.1x) | 7.9 (1.6x) | 6.6 (2.9x) | 5.3 (3.8x) | 0.47 | 0.83 | 1.13 | 1.48 |
| Synthesizer | 8.7 (1.1x) | 5.7 (1.2x) | 6.6 (2.9x) | 1.9 (1.4x) | 0.65 | 1.98 | 4.09 | 6.99 |
| BigBird | 7.4 (0.9x) | 3.9 (0.8x) | 2.7 (1.2x) | 1.5 (1.1x) | 0.77 | 1.49 | 2.18 | 2.88 |
| Linear Trans. | 9.1 (1.1x) | 9.3 (1.9x) | 8.6 (3.7x) | 7.8 (5.6x) | **0.37** | 0.57 | **0.80** | 1.03 |
| Performer | **9.5** (1.2x) | **9.4** (1.9x) | **8.7** (3.8x) | **8.0** (5.7x) | **0.37** | 0.59 | 0.82 | 1.06 |

Table 2: Benchmark results of all Xformer models with a consistent batch size of 32 across all models. We report relative speed increase/decrease in comparison with the vanilla Transformer in brackets besides the steps per second. Memory usage refers to per device memory usage across each TPU device. Benchmarks are run on 4x4 TPU V3 Chips. We report inference speeds in the supplementary material.

## 3.4 EFFICIENCY BENCHMARKS

In this section, we report efficiency metrics of our runs. For simplicity, we use the byte-level text classification benchmark and report run times and memory consumption of the sequence lengths $\{1K, 2K, 3K, 4K\}$. We use a batch size of 32 (1 example per core) for all runs and conduct experiments on 4x4 TPU V3 Chips. We emphasize that these runs are again largely conditioned on hardware and implementation details. For speed (steps per second), this is based on realistic training speed (i.e., including overheads such as io, batching, pipeline etc.) More details can be found in the appendix. Do note that the main point of these efficiency numbers is to provide a relative comparison. For follow-up work, it would be ideal to always rerun these numbers on a comparable hardware/setup. We report inference speeds in the supplementary material.

**Results on Train Speed** Table 2 reports our efficiency benchmarks on the xformer models. We note that low-rank and kernel-based models are generally the fastest. The overall fastest model is the Performer model (Choromanski et al., 2020), which is $5.7\times$ faster than Transformers on the $4k$ sequence length. Linformer (Wang et al., 2020) and Linear Transformers (Katharopoulos et al., 2020) come in a close second and are almost as fast as Performers (at $5.5\times$ to $5.6\times$ faster). Local Attention is also considerably fast. Based on our implementation, the slowest model is the Reformer model (Kitaev et al., 2020) that is about $80\%$ the speed of vanilla Transformer at $4K$ sequence lengths and half the speed at $1K$ sequence length.

**Results on Memory Consumption** The model with the smallest memory footprint in our benchmarks is the Linformer model, coming in at 0.99GB per TPU device as compared to 9.48GB per TPU device for the vanilla Transformers at $N = 4K$. That is about a 10x reduction in memory footprint. Similar to speed, Performers and Linear Transformers are also relatively compact and are almost as compact as Linformers. Other models (Local Attention, Reformers, BigBird, Synthesizers) are still less memory hungry compared to vanilla Transformers but are relatively less efficient (memory consumption wise) compared to Linformers, Performers, and Linear Transformers. We also notice that the memory consumption of models such as Linformer and Performer scales very well, with the memory usgae at $3K$ and $4K$ being approximately equal.

## 3.5 OVERALL RESULTS: NO ONE-SIZE-FITS-ALL

Based on our analysis, the best qualitative performance in terms of LRA score, i.e. integrated across all five tasks, is the BigBird model. While BigBird does not do extremely well on any individual task compared to other models, it has consistently good performance across all tasks. Performers and Linear Transformers have strong performance on some tasks but their average is lowered by the ListOps task.

Figure 3 shows the trade-off between qualitative performance (y-axis), model speed (x-axis), and memory footprint (size of the circles). While BigBird performs well, its speed is almost similar to the vanilla Transformer. In fact, based on our efficiency benchmarks in Table 2, it is slightly slower at shorter sequences (i.e., 1K-2K). On the other hand, a model like Local Attention is fast at the cost of lower quantitative performance. Among these models, the kernel-based variants, i.e., Performer, Linformer, and linear Transformer seem to be able to make a better trade-off in terms of speed and performance, while having reasonable memory usage. Overall, the models that lie on the pareto-optimal curve is BigBird and Performers.

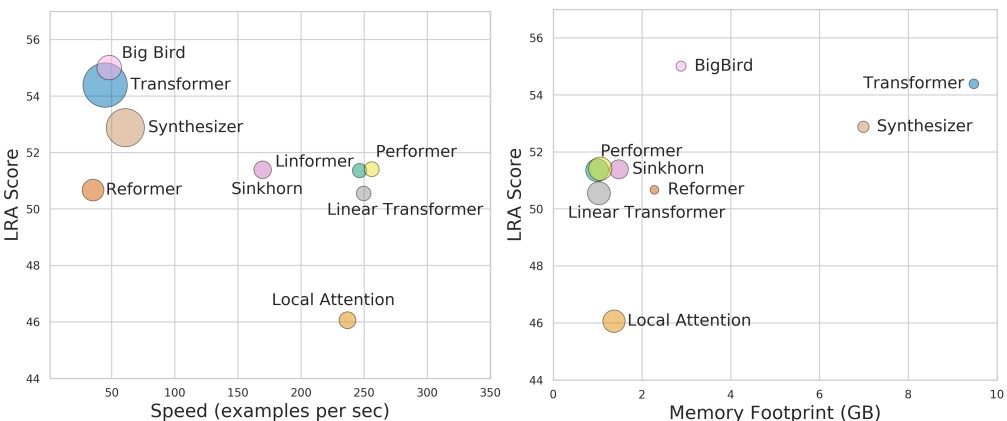

Figure 3: Trade-off between performance ($y$ axis) and resources ($x$ axis). On the left, circle size corresponds to memory footprint. On the right, it corresponds to examples per second.

## 4    RELATED WORK

### 4.1    EFFICIENT TRANSFORMERS

The pervasiveness of Transformer models, along with its well-known trait of being memory intensive, has spurred on a large number of innovations on this front. Early work in this area has typically considered a fixed pattern (local window) approach (Liu et al., 2018; Parmar et al., 2018). More advanced models have been proposed recently. Early work in this area has typically considered combinations of fixed patterns (Child et al., 2019; Ho et al., 2019; Beltagy et al., 2020; Zaheer et al., 2020) that learn sparse attention by considering combinations of fixed strides or local windows. There has been also interesting work in striving to learn these patterns patterns (Kitaev et al., 2020; Roy et al., 2020) using LSH hashing, clustering (Roy et al., 2020) and/or sorting (Tay et al., 2020b). The latest models are largely based on kernels (Katharopoulos et al., 2020; Choromanski et al., 2020) or low-rank approximations (Wang et al., 2020) which either treat the attention matrix as low-rank (using low-rank projections Wang et al. (2020)) or rewriting of the self-attention equation (Katharopoulos et al., 2020). For the sake of brevity, we refer interested readers to (Tay et al., 2020c) for a detailed survey of this line of research.

### 4.2    EXISTING BENCHMARKS

**Generative Modeling / Language Modeling**    This generative modeling task requires predicting the next character, word, or pixel and is a staple in xformer evaluations (Roy et al., 2020; Kitaev et al., 2020). However, it has been debated how much long-range signal such tasks actually encode (Rae & Razavi, 2020). LSTM language models augmented with attention have been shown to rarely attend beyond seven preceding words of context (Daniluk et al., 2017) and samples from LSTM language models are known to quickly devolve into generic text. On the other hand, recent models such as the Transformer-XL (Dai et al., 2019) have been observed to be sensitive to a context of around 900 tokens and samples from large-scale models (Radford et al., 2019) maintain a consistent theme over much longer sequences. Even such recent models, however, can be improved by limiting the range of attention (Rae & Razavi, 2020). In sum, while standard language modelling datasets contain *some* long-range signal, which is required to perform long-range coreference resolution, reasoning

with events, discourse understanding, etc. (Ruder et al., 2019) it seems to be overshadowed by the much stronger signal of short-term word co-occurrences and is thus difficult to evaluate.[4]

**Question Answering**    Another commonly used evaluation task is question answering (QA; Zaheer et al., 2020). Open-domain QA in particular typically requires the model to answer questions based on long contexts such as entire Wikipedia documents (Joshi et al., 2017; Kwiatkowski et al., 2019) or even books (Kočiský et al., 2018). Other datasets are explicitly designed to require multiple 'hops' of reasoning (Welbl et al., 2018; Yang et al., 2018). Successful approaches are often highly engineered, computationally expensive systems that require pre-training and a separate retrieval model (Lee et al., 2019; Guu et al., 2020).

**Natural Language Understanding / GLUE tasks**    Evaluation on natural language understanding (NLU) tasks is also common (Wang et al., 2020). Examples in most of these datasets such as MultiNLI (Williams et al., 2018) and SST (Socher et al., 2013) consist of single sentences and less than 100 tokens on average.

## 5    CONCLUSION

We proposed Long Range Arena (LRA), a new benchmark for evaluating progress on efficient Transformer research. Our new benchmark is challenging and probes at model capabilities in dealing with diverse data types and structures such as text, mathematics, and visual data. Our benchmark comprises of tasks ranging from $1K$ to $16K$ tokens. For the first time, we conduct an extensive side-by-side comparison of ten recently proposed efficient Transformer models. The experimental results show that these tasks are very challenging even for long-range Transformer models. The overall results show that there is no one-size-fits-all solution and trade-offs have to be made in terms of model quality and speed/memory. We plan to open source our code and benchmarks to facilitate future benchmarking, research and model development.

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

# A APPENDIX

## A.1 LRA TASKS

This section describes the details and hyperparameters of each task. We also plan to release the configuration files along with the implementation of the models and benchmarks, that can be used to reproduce the results reported in the paper.

### A.1.1 LISTOPS

Following the generation steps in (Nangia & Bowman, 2018), we generate our own long version of this task. We use a sequence length of $2k$ for this task. All our xformer models have an embedding dimension of $512$, 8 heads, 6 layers and a feed-forward dimensions of $2048$. We train all models for $5K$ steps. The [CLS] token is used and mapped into a 10 class Softmax layer for classification.

### A.1.2 BYTE-LEVEL DOCUMENT CLASSIFICATION

We use the IMDb reviews dataset (Maas et al., 2011) and a sequence length of $\{1K, 2K, 3K, 4K\}$ tokens for all models. We pick the best results across these four sequence lengths. We use a [cls] token for prediction. All the [cls] tokens from xformer encoders are passed into a two layered MLP with ReLU activations. The MLP emits a 2-class logits for binary classification. We optimize the softmax cross entropy loss function. All xformer models are parameterized by the same number of layers, heads and hidden dimensions, namely 8 heads, $512$ hidden dimensions and $d = 2048$ for positional FFN layers. We use 6 layers for all xformers. The learning rate is $0.05$ with weight decay of $0.1$. We use Adam with warmup. All models are trained for $20K$ steps and a batch size of $32$.

### A.1.3 BYTE-LEVEL DOCUMENT MATCHING

We use the ACL anthology network for a related article matching task. We use a sequence length of $4K$ per document ($8K$ tokens in total for two sequences). The two encoders share parameters. Similar to document classification, we use the [cls] token from xformer encoders. Let $X_1$ be the [cls] token embedding from document 1 and $X_2$ be the [cls] token embedding from document 2, the final score is computed via:

$$Y = \mathrm{MLP}([X_1, X_2, X_1 * X_2, X_1 - X_2])\tag{1}$$

where MLP(.) is a two layered MLP with relu activation functions. In lieu of the much longer sequence length, we use a batch size of $32$, embedding dimension of $128$, 4 heads, a FFN dimension of $512$ and 4 layers. Model is trained with Adam for $5K$ steps with a learning rate of $0.5$.

## A.2 IMAGE CLASSIFICATION

We use the gray-scaled (single channel) CIFAR10 as the image classification dataset, with 10 classes. The resolution of input images is $32 \times 32$ and after flattening the input images, we feed our xformer encoders with a sequence of 1024 pixels. Similar to our other classification tasks, there is a classifier head on top of the xformer encoder, consisting of a two-layer MLP with ReLU activation. Softmax cross-entropy has been used for optimizing the parameters of the models. We trained our models for 200 epochs and have done extensive sweeps over different hyper-parameters and found the following values leading to the best average performance across all xformers: 3 layers, 4 heads, 128 as the hidden dimensions of FFN blocks, 64 as the query/key/value hidden dimensions, and finally the learning rate of $0.01$.

### A.2.1 GENERALIZATION GAP

For the image classification benchmark, in Section 3, we mentioned that most of the models struggle generalizing to the test set. Table 3 presents the train and test accuracy for different models and for almost all these models, the gap between the two scores is considerably high.

While this task can be simple to solve for convectional models (e.g., accuracy of wide-resnet on gray-scale CIFAR10 with no data augmentation is 89.21) it is rather difficult for Transformer-based

| Model | test accuracy | train accuracy |
|---|---|---|
| Transformer | 42.44 | 69.45 |
| Local Attention | 41.46 | 63.19 |
| Sparse Trans. | **44.24** | 66.74 |
| Longformer | 42.22 | 71.65 |
| Linformer | 38.56 | 97.23 |
| Reformer | 38.07 | 68.45 |
| Sinkhorn Trans. | 41.23 | 69.21 |
| Synthesizer | 41.61 | **97.31** |
| BigBird | 40.83 | 71.49 |
| Linear Trans. | 42.34 | 65.61 |
| Performer | 42.77 | 73.90 |

Table 3: Test and train accuracy of different models on Image Classification task.

models with this setup. Naturally, one can find ways to improve the performance with a different setup. For instance, in our setup, models are not informed about the ordinality of pixel intensities and consume them as independent symbols. We observed that learning embedding that reflects this property is rather hard for most of these models (Figure ). If we simply replace the embedding layer with a CNN stem, we see imitate boost in the performance (e.g. replacing the embedding layer of a vanilla Transformer with a convectional stem, with $3 \times 3$ kernel, we get accuracy of 75.32).

Another modification that can lead to better performance is to incorporate spatial representation that are translation invariant in Transformer models (e.g., adding 2D relative positional embedding to a vanilla transformer, we get accuracy of 61.72). However, adding these sorts of changes make the setup digress from the original point of this task in our benchmark.

### A.2.2 VISUALIZATIONS OF LEANED EMBEDDING BY A VANILLA TRANSFORMER

Figure 4 presents visualizations for the pixel intensity and positional embedding that a vanilla transformer model learns for the image classification task, on the gray-scaled CIFAR10 detest.

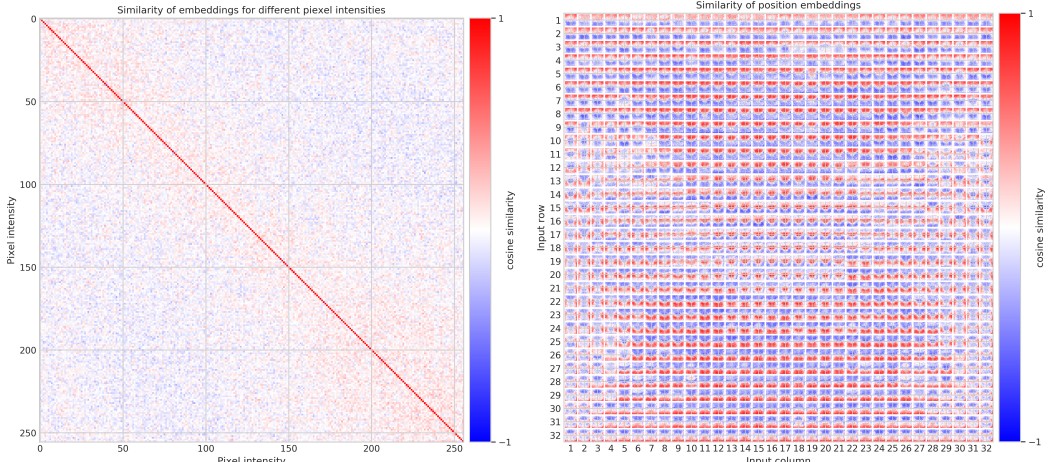

Figure 4: **Left:** The cosine similarity between the embedding learned for each pixel intensity. **Right:** Each tile shows the cosine similarity between the position embedding of the pixel with the indicated row and column and the position embeddings of all other pixels.

On the left, we can see the pairwise similarity of learned embeddings for pixel intensities. Although there is a higher similarity for close pixel values, the patterns from these learned embeddings do not perfectly reflect the ordinality of the pixel intensities. On the right, we can see the pairwise similarity of positional embeddings for different input positions. We can see that the lower the

distance between two pixels is, the more similar are their learned positional embeddings. However, the spatial closeness in $y$ axis is more preserved in the learned embedding than the distances in the $x$ axis.

## A.3 PATHFINDER

Pathfinder task probes the ability of models to detect long range spatial dependencies between input features. To solve the task, a model requires to identify the target contour and trace it from one end to the other. Although Pathfinder is visually a simple task, it has been show that the clutter and variations in path shape makes the task difficult for CNN models (Linsley et al., 2018; Kim* et al., 2020).

The Pathfinder task is a binary classification task and the resolution of input images is $32 \times 32$. Similar to image classification task, we feed our xformer encoders with a sequence of 1024 pixels after flattening the input images. The classifier head on top of the xformer encoder is also a two-layer MLP with ReLU activation and we use Softmax cross-entropy loss for the optimization. We trained our models for 200 epochs. The hyper-parameters used for the xformer model are as follow: 4 layers, 8 heads, 128 as the hidden dimensions of FFN blocks, 128 as the query/key/value hidden dimensions, and the learning rate of 0.01.

### A.3.1 VISUALIZATION OF THE ATTENTION MAPS FROM A VANILLA TRANSFORMER

Given that transformers have many units with global receptive field, they have better potential for solving the task, compared to models with local receptive fields. Figure 5 shows the attention distributions for a set of examples given on token (CLS token) as the query. We can see that the attention module collects information from different positions in input to be able to trace the target path.

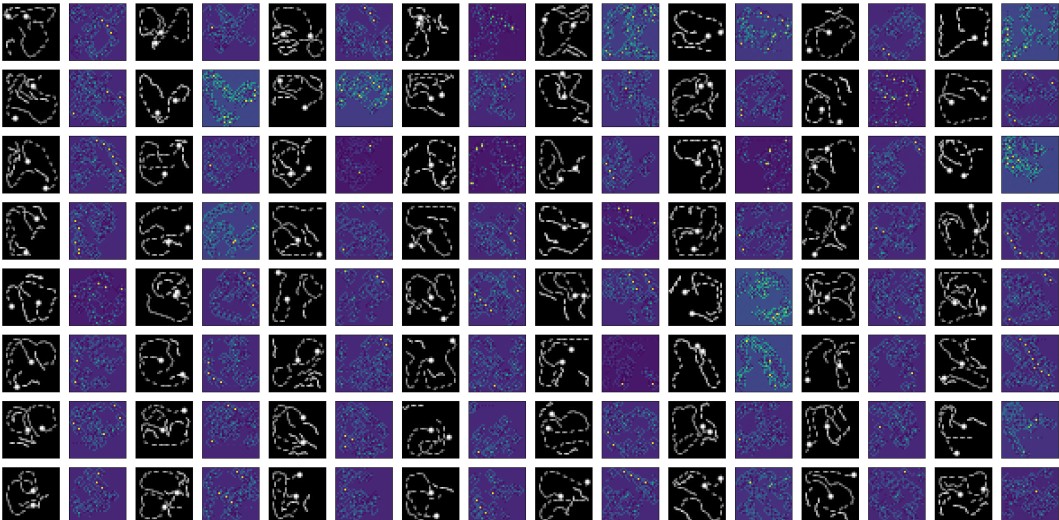

Figure 5: Attention map for different examples from the Pathfinder task. Each map presents the attention distribution, given the CLS token at the final layer as the query, averaged across all heads in a vanilla Transformer model. Note that for visualization, we use attention-rollout (Abnar & Zuidema, 2020) for more precise input attribution.

We have also included a Pathfinder-X in LRA, which is similar to Pathfinder, but inputs are in higher resolutions, i.e. longer input sequences. On Pathfinder-X, we have tried two setups for training our models, first training models from scratch, second evaluating models that are trained on Pathfinder. In both cases, we found out none of the models are able to deal with/generalize to 16K input length.

## B    MODELS AND IMPLEMENTATION

This section describes the details of our implementation. The code is primarily written in JAX and FLAX. In this section, we note specific details about certain implementations of models. We plan to release hyperparameters in a form of readme or script later.

### B.1    SPECIAL CASES OF OUR IMPLEMENTATION

This section describes several special cases in our implementation details. The diverse suite of Transformers come with a plethora of hardware constraints and implementation details. To succeed, a Transformer model needs to also 'win' the hardware lottery (Hooker, 2020), i.e., having readily supported ops, kernels or accelerator support to take advantage of its technical design. This section discusses some of the trade-offs and edge cases that make comparison of several models challenging. In the end, we argue that simplicity is a virtue and not requiring any special support is a positive thing for an efficient Transformer model.

**On CUDA kernels**    CUDA kernels are cumbersome and are specific to GPU hardware, making it difficult to implement or use on TPU pods. Generally, these are considered to be undesirable and inconvenient in practical applications. Hence, Sparse Transformer and Longformer are implemented with **equivalent** implementations to emulate for performance. This is by applying an equivalent mask. For this reason, we do not benchmark Sparse Transformer and Longformer for speed.

**Reformer's Implementation**    Having optimized ops to support many of Reformer's functionality is crucial. Hence, Reformer is implemented slightly differently from other Transformer models. Instead of computing tensors with batch size dimensions $B$ and head dimensions $H$, (i.e., $B \times H \times N \times d$), we compute the attention function for tensors of $N \times d$ dimensions. After which, we parallelize this function via VMAP over the batch and head dimensions.

## C    INFERENCE SPEED BENCHMARKING

In section 3.4, we presented the comparison between the speed of different models at the training time. This section provides the results of a similar comparison, but at the evaluation/inference time.

| Model | 1K | 2K | 3K | 4K |
|---|---|---|---|---|
| Transformers | 114 | 64 | 28 | 16 |
| Local Attention | 108 (0.47x) | 110 (1.71x) | 108 (3.86x) | **111** (6.93x) |
| Reformer | 54 (0.47x) | 27 (0.42x) | 18 (0.64x) | 13 (0.81x) |
| Synthesizer | 111 (0.97x) | 106 (1.65x) | 55 (1.96x) | 31 (1.94x) |
| Sinkhorn Transformer | 111 (0.97x) | 108 (1.69x) | 110 (3.92x) | 100 (6.25x) |
| Linformer | 112 (0.98x) | **111** (1.73x) | 109 (3.89x) | 110 (6.88x) |
| BigBird | 70 (0.61x) | 34 (0.53x) | 23 (0.82x) | 17 (1.06x) |
| Linear Transformers | 108 (0.95x) | **111** (1.73x) | 109 (3.89x) | **111** (6.94x) |
| Performer | **116** (1.02x) | 110 (1.72x) | **116** (4.14x) | 110 (6.89x) |

Table 4: Speed (Steps per second) on running inference. Benchmarked on 4x4 TPU V3 chips with a batch size of 32 (1 example per core). Results are computed on Xformers with 4 layers, 8 heads, 128 hidden size. Similar to training, we include realistic evaluation time which includes pipeline ops, and batching into the overall time.

**Inference results**    Overall, the relative results and trends on inference is not too different from training (shown in Table 2 of the main paper). Similarly, Performer, Linear Transformers, Linformer and Local Attention remains to be very strong in terms of inference speed. It is worth to also note that all xformer variants are slower than the vanilla Transformer at $1K$ length. This is unlike training, where most xformers are performing at $\approx 1.1$x speed of vanilla Transformers.

# D    CONVERGENCE ANALYSIS

We analyze the convergence quality of these xformer models. We report the steps to $N\%$ accuracy on the Image benchmark. Given that the final val accuracy is about $38\%$ to $40\%$ for most models, we report the time for models take to reach $30\%$ and $35\%$ as an estimate of how fast these models converge[5].

| Model | $N = 30\%$ | $N = 35\%$ |
|---|---|---|
| Transformer | 2048 | 3920 |
| Local Attention | 1425 | 3512 |
| Sparse Transformer | 1050 | 3325 |
| Reformer | 1720 | 6650 |
| Linformer | **452** | **875** |
| Longformer | 713 | 1575 |
| Sinkhorn Transformer | 875 | 1575 |
| Synthesizer | 2975 | 3325 |
| Linear Transformer | 2186 | 3500 |
| BigBird | 1225 | 2462 |
| Performer | 1400 | 2625 |

Table 5: Number of training steps to reach $N\%$ validation accuracy where $N = \{30, 35\}$.

**Results**  The Linformer model converges the fastest, followed by Longformer, Sinkhorn Transformers and then BigBird. The vanilla Transformer is a little on the slow side of convergence but is still faster than Reformer, the slowest model to converge. Notably, the convergence speed is not a signal of how well it would finally perform as noted in the results in Table 1.

---

[5]accuracy tends to

# E  ARCHIVAL SNAPSHOTS OF THE RUNS

This section reports the archival snapshots and change log of the leaderboard and comparisons of the different model. As we tune hyperparameters and fix certain issues, we will update the snapshot of the model comparison here. Future versions may also include other models that we benchmark and compare. For easy comparisons in future papers, please quote the version number of the leaderboard. We will continiously update this section in the appendix if there are any futhur updates on the existing models.

## E.1  VERSION 1: ICLR CAMERA READY.

Version 1 is active as of March 2021. The main changes is that we reran all models for the ListOps task. We train models for longer this time round for up to 10K steps.

| Model | ListOps | Text | Retrieval | Image | Pathfinder | Path-X | Avg |
|---|---|---|---|---|---|---|---|
| Chance | 10.00 | 50.00 | 50.00 | 10.00 | 50.00 | 50.00 | 44.00 |
| Transformer | 36.38 | 64.27 | 57.46 | 42.44 | 71.40 | FAIL | 54.39 |
| Local Attention | 15.95 | 52.98 | 53.39 | 41.46 | 66.63 | FAIL | 46.08 |
| Sparse Trans. | 35.78 | 63.58 | **59.59** | **44.24** | 71.71 | FAIL | 54.98 |
| Longformer | 36.03 | 62.85 | 56.89 | 42.22 | 69.71 | FAIL | 53.54 |
| Linformer | 35.49 | 53.94 | 52.27 | 38.56 | 76.34 | FAIL | 51.32 |
| Reformer | 36.30 | 56.10 | 53.40 | 38.07 | 68.50 | FAIL | 50.47 |
| Sinkhorn Trans. | 34.20 | 61.20 | 53.83 | 41.23 | 67.45 | FAIL | 52.78 |
| Synthesizer | 36.50 | 61.68 | 54.67 | 41.61 | 69.45 | FAIL | 52.78 |
| BigBird | **37.08** | 64.02 | 59.29 | 40.83 | 74.87 | FAIL | **55.22** |
| Linear Trans. | 17.15 | **65.90** | 53.09 | 42.34 | 75.30 | FAIL | 50.76 |
| Performer | 36.00 | 65.40 | 53.82 | 42.77 | **77.05** | FAIL | 55.01 |
| Task Avg (Std) | 32 (7.9) | 61 (4.6) | 55 (2.6) | 41 (1.8) | 72 (3.7) | FAIL | 52 (2.4) |

Table 6: Experimental results on Long-Range Arena benchmark. Best model is in boldface and second best is underlined. All models do not learn anything on Path-X task, contrary to the Pathfinder task and this is denoted by FAIL. This shows that increasing the sequence length can cause seriously difficulties for model training. We leave Path-X on this benchmark for future challengers but do not include it on the Average score as it has no impact on relative performance.

### E.1.1  NOTABLE CHANGES

The main outcome on the new leaderboard is as such:

- The relative results on Listops largely remain unchanged for most models. However, models such as Performer and Linformer managed to get comparable results to the other models with more training steps.

- Performer is now the 2nd best model. BigBird is still the leading model.

## E.2  VERSION 0: ORIGINAL RESULTS TO ICLR

Version 0 ran from September 2020 to March 15th 2021.

| Model | ListOps | Text | Retrieval | Image | Pathfinder | Path-X | Avg |
|---|---|---|---|---|---|---|---|
| Chance | 10.00 | 50.00 | 50.00 | 10.00 | 50.00 | 50.00 | 44.00 |
| Transformer | 36.37 | 64.27 | 57.46 | 42.44 | 71.40 | FAIL | 54.39 |
| Local Attention | 15.82 | 52.98 | 53.39 | 41.46 | 66.63 | FAIL | 46.06 |
| Sparse Trans. | 17.07 | 63.58 | **59.59** | **44.24** | 71.71 | FAIL | 51.24 |
| Longformer | 35.63 | 62.85 | 56.89 | 42.22 | 69.71 | FAIL | 53.46 |
| Linformer | 35.70 | 53.94 | 52.27 | 38.56 | 76.34 | FAIL | 51.36 |
| Reformer | **37.27** | 56.10 | 53.40 | 38.07 | 68.50 | FAIL | 50.67 |
| Sinkhorn Trans. | 33.67 | 61.20 | 53.83 | 41.23 | 67.45 | FAIL | 51.39 |
| Synthesizer | 36.99 | 61.68 | 54.67 | 41.61 | 69.45 | FAIL | 52.88 |
| BigBird | 36.05 | 64.02 | 59.29 | 40.83 | 74.87 | FAIL | **55.01** |
| Linear Trans. | 16.13 | **65.90** | 53.09 | 42.34 | 75.30 | FAIL | 50.55 |
| Performer | 18.01 | 65.40 | 53.82 | 42.77 | **77.05** | FAIL | 51.41 |
| Task Avg (Std) | 29 (9.7) | 61 (4.6) | 55 (2.6) | 41 (1.8) | 72 (3.7) | FAIL | 52 (2.4) |

Table 7: Experimental results on Long-Range Arena benchmark. Best model is in boldface and second best is underlined. All models do not learn anything on Path-X task, contrary to the Pathfinder task and this is denoted by FAIL. This shows that increasing the sequence length can cause seriously difficulties for model training. We leave Path-X on this benchmark for future challengers but do not include it on the Average score as it has no impact on relative performance.

