# OpenReview forum: "Long Range Arena : A Benchmark for Efficient Transformers "
_ICLR.cc/2021/Conference — ICLR 2021 Poster_

### Official Review · AnonReviewer2 · 2020-10-22
**An important, well-executed benchmark (though consisting of non-realistic tasks) for evaluating transformers on long-range tasks.**

**Rating:** 7
**Confidence:** 4

**Review:**

This paper presents LongRangeArena (LRA), a new benchmark for evaluating models such as Transformers on tasks that require long-range processing. The authors present a suite of 6 tasks, each requiring access to tokens that are hundreds and even thousands of tokens apart from the target token. Their tasks share a set of important properties such as being challenging and yet simple (though not realistic, see below). The authors make a thorough comparison between 9 models designed to support efficient processing of long inputs, and show nice tradeoffs between performance, speed and memory usage. Overall, this paper makes an important contribution and despite some limitations should be accepted to ICLR.

The paper tackles an important problem: evaluating long-range transformers. There have been multiple efforts recently to design models that scale better than the quadratic time and space complexity of vanilla transformers to long sequences. Despite the large number of works, there hasn't been a standard benchmark to compare them against each other, which makes it harder to evaluate the quality of each of them. This paper addresses this problem, and presents a relatively good solution: a single, diverse benchmark that requires long-sequence processing. The new benchmark is likely to promote research on this important topic.

My main concern with this benchmark is that it doesn't involve any realistic task. The proposed tasks are either synthetic (ListOPs) or made artificially hard by forcing byte-level processing or flattening images to vectors. This means that there is no real value in performing well on these tasks, other than being a potential proxy to other tasks with similar characteristics, assuming such exist. I think relating this benchmark to real tasks would have made it much stronger. I realize that the authors made an attempt to control for different variables such as structure, but I am still not convinced that models that do well on this benchmark will be useful for anything real.

Other comments:
1. The paper would benefit from discussing the differences between the different models. A very minimal discussion is given on the last page, with a pointer to a survey paper. While I realize this is not the main focus of the paper, the different models are an important part of it, and without such even basic comparison, it is hard to appreciate the analysis presented in the results section (e.g., the last sentence on the Results on ListOps).
2. The requirement that the tasks should allow for "probing diverse aspects" of the models is somewhat under-delivered. The authors discuss some probing aspects of the first 3 tasks, but do not come back to these later in the paper, which was a bit disappointing.


Minor:
1. Adding random/majority baselines to table 1 would make it easier to appreciate the results
2. Results on ListOps: "the inductive bias of the xformer *model* plays " -> *models*

---

> ### Comment · AnonReviewer2 · 2020-11-15
> **Training or inference speed?**
>
> Dear authors,
>
> I wasn't able to find in the paper whether the numbers reported in Table 2 reflect training or inference speed. Am I missing something? It would be nice to see both actually.

---

> ### Author Response · Authors · 2020-11-18
> **Response to review**
>
> Thanks so much for the insightful and constructive feedback! We are appreciative of your valuable time spent to provide this high quality feedback.
>
> This review brings up an important point about synthetic vs realistic tasks. This is something that we have deeply considered when designing the benchmark. Let me share our thoughts and considerations pertaining to this topic. First and foremost, we see the tasks in our benchmark as a means to better understand and analyze this class of models. Amongst which, the ListOps, Pathfinder and Path-X are completely synthetic tasks while Text, Retrieval and Image comes from realistic distributions.
>
> Perhaps the key question here is how much we do expect these tasks to translate to performance on real world tasks. We wish for our benchmark to be used as a toolkit for analysis and a final score is not authoritative (and does not imply better performance on real world tasks). We feel that asking questions such as "why does Reformer do so well on hierarchical data" and "why does Performer do the best on spatial reasoning", may help the community design better architectures. While the overall score may or may not mean much to real world tasks, we believe that, moving forward, there is considerable merit.
>
> Notably, there are already a myriad of real world tasks that people compare their (often pretrained) transformers on. However, it is non-trivial to probe and test long-range models in many of these tasks (as discussed in our paper). Our goal was to design a complementary benchmark that researchers can compare their models with. The goal was to provide a clearer picture and we hope this complementary benchmark will give researchers a better view, perspective and insight to the performance of these models under different scenarios.
>
> Regarding the point of discussion (of Transformer models), we will include more details of these different xformer models to the revised paper. Thanks for the suggestion!
>
> Regarding the point on probing diverse aspects, this refers to the tasks/data having very different underlying structures such as hierarchical or spatial structures. For example, ListOps probes for hierarchical reasoning abilities of the xformer model. As such, we can infer that a good model on ListOps is good at hierarchically structured data. We discuss this (briefly) in our experimental results (for example, we inferred that models such as Performer, Linformer are possibly not effective on hierarchically structured data. We agree that this could be made clearer and we will make this modification in the revised version! Thanks for the suggestion!
>
> Regarding the steps per second, this refers to training speed. We will also report the inference speed in the revised version.
>
> Once again, thanks for the great review and taking the time to review our paper!
>
> Note: We will update the revised paper by the 24th November deadline.

---

> > ### Comment · AnonReviewer2 · 2020-11-18
> > **Response to response**
> >
> > Thank you for the detailed response. I agree that the benchmark is useful as it is, and am generally supportive of the paper, but still think relating it to a more realistic setting would make the paper much stronger. This is obviously a question for future work at this point.

---

### Official Review · AnonReviewer1 · 2020-10-26
**Interesting and thorough analysis, unclear whether this is a useful benchmark**

**Rating:** 7
**Confidence:** 4

**Review:**

*Summary*: This paper proposes a new benchmarks for the host of recently-proposed transformer variants focused on efficiency and scaling to longer sequence lengths (xformers). The authors reimplement and study the performance of 10 xformers on their benchmark. Furthermore, the authors conduct a study of the memory consumption and speed of the models on their text classification benchmark.

*Strengths*: Detailed and thoughtful comparison of 10 models aimed at more-or-less alleviating the same problems. The tasks span a wide range of sequential data modalities.

*Weaknesses*: It’s not entirely clear to me that LRA is best-positioned as a benchmark, rather than an analysis tool; the authors themselves seem to also note this (“Hence, the results provided in this paper are not meant to be a final authoritative document on which xformer is the best”). This toolkit seems more useful to me as an analysis tool---the choice of tasks itself also reflects the benchmark’s values. For instance, a NLP researcher might care more about performance of ListOps, since it’s possible that these results have more relevance for the type of structure found in natural language.

Instead of seeking to rank models (calling LRA a “benchmark” and having an “average” LRA score directly feeds into this), I’d like to see this toolkit shift toward being more customizable for individual user questions and values; similarly, this paper might take a more analytical approach to empirically studying the performance of these 10 models on this starter set of tasks, versus trying to rank them.

Despite this lack of clarity of goals, I think that this toolkit and study offer useful contributions.

*Recommendation*: 7 . While it remains unclear to me that this paper has a useful benchmark contribution, the analysis of existing models is valuable. Furthermore, the observation that inherent tradeoffs in performance and speed make no model the one-size-fits-all option is important; in light of this, I’d like to see the authors move towards making their toolkit better for determining what the “right” option is for a given user’s use case.

Comments and Questions:

Figure 2: y axis label should be “Span”, not “Apan”

Some abstraction is taken with respect to hyperparameters---a single set of hyperparameters is used across all models. Do you have a sense of how much this can potentially impact performance? For instance, for a single task, comparing results when you use this single set of hyperparameters vs individually tuning models for the task.

Do you think that future developers of xformers should “hillcimb” on LRA?

---

> ### Author Response · Authors · 2020-11-18
> **Response to review**
>
> Thanks for the insightful comments and feedback, along with taking the time to review our paper! We are greatly appreciative of the extremely high quality feedback.
>
> Honestly, we think that this is a great review. Your review has made us think more critically about the benchmark and even more so about the philosophy behind it. We appreciate this a lot. Thank you for this.
>
> Many great points were raised and the main ones were pertaining to the positioning of LRA as a benchmark. In the paper, we were very cautious to not promote a culture of explicitly ranking these xformer models. Our goal was to evaluate and analyze these classes of models. In the end, the aggregate scores, performance-compute trade-off graphs were all tools in aid of researchers who want to explore these models for their use cases. As noted, this direction was implicitly conveyed throughout our paper, i.e., emphasizing that these results tend to change over time with hyperparameter tuning and about how we downplay any form of “ranking” culture by emphasizing that there is no ultimate xformer model. We want to focus on making systematic and meaningful comparisons, instead of a winner-take-all culture where all it matters is to get a better number on this leaderboard. We think that this is very aligned with what you have mentioned.
>
> The choice of naming LRA as a benchmark simply stems from the fact that we want to introduce LRA as a set of tasks that one can “benchmark” (analyze) their model on. This brings us to the question about “hillclimbing”, We think that the current setup of LRA makes it difficult for incremental hillclimbing. For starters, unlike other benchmarks with a multitude of variables and moving parts such as contextual embeddings, pretraining and possibly many other architectural tricks to improve performance etc, we’re restricting advancements to only the efficient attention mechanism in a controlled setting. We’ll also emphasize the need for equal parameterization and all comparisons made should be apples-to-apples. In other words, we’re focusing on ablative experiments that validate certain research hypotheses instead of focusing on an absolute final number. We believe this provides an interesting “inductive bias” that encourages hypothesis driven research instead of hillclimbing or SOTA chasing.
>
> Researchers can use our benchmark in a myriad of ways. They may show that they design a model that does very well on the compute-performance curve (being pareto-optimal). They may design a model that solves or does one of the task very well (but still lack in overall LRA score). They may provide analysis on why kernel models do better or worse on a certain data type/modality. They may provide insight on why Path-X is much harder than Pathfinder. I think there are numerous research questions that our benchmark poses to the community.
>
> We think that you have made a very good point about providing guidance to users on how they might interpret our results and tailor them for their use case. We will add a section on this in our paper.  Finally, regarding hyperparameters, we did a comprehensive sweep for the base Transformer to ensure they were giving reasonable good results to begin with. Within this “reasonable” range and excluding hparams that turned out to be horrible, the variance is not high. Because of this, we believe that the variance of every model would also be similar. We leave extensive hparams tuning of model specific hparams to future work.
>
> Once again, thanks for the great review!
>
> Note: We will update the revised paper by the 24th November deadline for revisions.

---

> > ### Comment · AnonReviewer1 · 2020-11-20
> > **Thanks for your response**
> >
> > Thanks for your response, especially in clarifying the polysemous use of "benchmark"---this hadn't crossed my mind before, and I now agree that it's an appropriate title. Looking forward to seeing the revision!

---

### Official Review · AnonReviewer3 · 2020-10-28
**Not a perfect benchmark, but still useful**

**Rating:** 7
**Confidence:** 4

**Review:**

This paper proposes benchmark tasks for the new crop of efficient Transformer models, an evaluation suite which the authors dub the "Long Range Arena." These tasks are selected to require hierarchical modeling of long-distance dependencies, and specifically to *exclude* pre-training or a need for pre-trained models (making many of the NLP tasks which motivate these papers unsuitable).  Many of the tasks have a somewhat artificial flavor: purely artificial (ListOps), treating images as sequences of pixels, or byte-level classification rather than word level or word piece level. Regardless, the paper shows interesting variation in the tasks across the different models, particularly on ListOps. BigBird seems to work the best out of the box, but the authors caution against reading too much into this, as they could not extensively tune hyperparameters on each of these models.

I think this paper addresses a desperate need in the literature, which is a way of making sense of all these Transformer variants that have been proposed. In this sense, I like what it's doing. My main reservations concern the somewhat artificial nature of the tasks and the generalizability of the results.

NATURE OF THE TASKS: This paper is probably wise to not conflate distinctions in architecture with distinctions in pre-training, and exploring both of these factors is too much and too complex for a single piece of work.  Papers such as the Longformer have primarily evaluated on NLP tasks like coreference and question answering; however, the "long-range" abilities on these datasets only add a few percentage points on the given metrics, and they rely on pre-training to be successful. However, I feel like the distribution of long-range effects in natural language is very different than the distribution of effects here: you have effects like entity coreference that aren't captured well by these tasks (these don't matter much for text classification and are absent from ListOps and the image tasks). So I'm not fully convinced that a model that does better on this benchmark suite would naturally do better on problems we really care about.

The tasks here do feature some nice effects: explicit hierarchy (ListOps), local composition (the byte-level tasks), and regular long-range correspondences (the image tasks). It's not clear to me how critical this last category is, as the 2D layout is an inductive bias that would almost be built into the architecture, but that's a separate discussion. However, the point remains that this benchmark does make choices about what is and isn't important, *particular* when you average across the datasets (ascribing each one equal weight), and I can imagine another benchmark being proposed focusing on a different set of effects. So I'm not sure how to judge this benchmark suite versus others that might exist.

RESULTS: This leads to my second point, about the results. I do like how the paper couches the results, explains the findings, and avoids overclaiming. I agree with the authors that, with some tuning, many of these approaches could possibly substantially better at various of these tasks. We might also see that some of these approaches are fundamentally not well-suited to these tasks.  But again, given the somewhat artificial nature of the tasks, what are we likely to actually learn? I foresee a future with ~10 papers hillclimbing on this benchmark, and I'm not convinced that those architectures will be any better than these when applied to new settings, LM pre-training, etc.

That said, the differences on ListOps are pretty interesting.  I wonder if Path-X has fundamental identifiability issues due to its scale: for example, there are now more geometries the model can learn from the sequence, and possibly there are somehow just too many of these for learning to effectively figure out which is correct. (I don't have a more formal argument, but it feels like a case of exponential numbers of hypotheses outstripping the size of the dataset as things scale up.)

All this said, I tend to come down positive on this paper: I think it brings some clarity to this space and will be a very useful starting point for others in the literature. And I don't think this meta-benchmark is necessarily more flawed than others like GLUE.  But nor do I think it's perfect.

---

> ### Author Response · Authors · 2020-11-18
> **Response to Review**
>
> Thanks for the insightful comments and feedback, along with taking the time to review our paper! We are greatly appreciative of the extremely high quality feedback.
>
> We agree, appreciate and align with many of the thoughts that you have. In the design and process of this work, we have deeply thought about many of these factors. The first point pertains to “how does one interpret” the results of this benchmark and how well would they translate to the problems we do care about. In general, we feel that the results should be interpreted as they are and one should not overly interpret them. For example, ListOps is not meant to be a substitute for the ability of a model to perform NLP tasks. Of course, a practitioner may find that a particular LRA task correlates with a specific quality that they desire in their problem space. Each benchmark isolates a single aspect that we want to evaluate a model on. We also hope that we do not imply that an xformer that does well on our LRA benchmark will be the best Transformer for all tasks, real or otherwise.
>
> We like the point that you have made about benchmarks being an “inductive bias” of what is important and what is not important. It is true that we present an aggregate score of all the tasks, seemingly implying that a model has to do reasonably well on all tasks to achieve a good overall score. Since our reply is “on the record” on openreview, we also like to emphasize that researchers should be free to use the benchmark to validate a certain research hypothesis (e.g., a better xformer for spatial reasoning) which makes the overall LRA score less important if they can do extremely well on one task (or validate certain research hypothesis).
>
> This brings us to the point about hillclimbing. [Note:this is also repeated as a reply to anonreviewer 1] We think the current setup of LRA makes it difficult for incremental hillclimbing. For starters, unlike other benchmarks with a multitude of variables such as contextual embeddings, pretraining and possibly many other architectural tricks to improve performance etc, we’re restricting advancements to only the efficient attention mechanism in a controlled setting. We’ll also emphasize the need for equal parameterization and all comparisons made should be apples-to-apples. In other words, we’re focusing on ablative experiments that validate certain research hypotheses instead of focusing on an absolute final number. We believe this provides an interesting “inductive bias” that encourages hypothesis driven research instead of hillclimbing or SOTA chasing.
>
> The point you have brought up about Path-X is indeed interesting that encourages future research on long sequences. The code used to generate these will be made open source so researchers can validate this hypothesis with a larger dataset. We initially thought that this was a potential problem in the Softmax mechanism in the self-attention inductive bias and perhaps interesting follow up work.
>
> Once again, many thanks for the detailed and well thought out review.

---

### Official Review · AnonReviewer4 · 2020-10-28
**Benchmarking efficient transformers**

**Rating:** 6
**Confidence:** 4

**Review:**

This paper proposes a suite of long sequence processing tasks to benchmark efficient transformer variants in terms of their accuracy/speed tradeoff. The benchmark tasks are constructed following six desiderta: generality, simplicity, challenging, long inputs, probing diverse aspects, and accessibility, resulting in six tasks mostly in the form of sequence classification where the sequences range from 1k to 16k tokens. Ten existing efficient transformer models are evaluated on these tasks and their accuracy/speed tradeoffs are compared on a relatively fair basis.

Pros:
1. This work is well motivated and timed. Given the attention efficient transformer variants recently attracted, it is important to have a common benchmark for comparability of results.
2. The diversity of the proposed tasks (in terms of types of tasks and modalities covered) enables a full grasp of the performance among different types of tasks, especially exposing the weaknesses. For example, it is interesting to see that kernel-based methods fail on tasks requiring hierarchical structures.
3. This benchmark is accessible for academia : from table 2 it takes at most 10G GPU memory for a normal transformer.

Cons:
1. For generality this suite of tasks only considered transformer encoders, but not the autoregressive decoding process, so the tasks here are mostly sequence classification tasks only. I think it is a limitation of the benchmark here. For example, all efficient transformers fail on path-x, but that don't mean that path-x is not useful. It is the limitation of the efficient transformer method itself if it does not support efficient autoregressive decoding.
2. While the benchmark proposed here considers image and text, it'd be interesting to add audio processing as well, such as hot word detection.
3. Pro 3 might be a downside of this benchmark as well, the tasks here can be mostly solved by a normal transformer while one reason we want efficient transformer is to apply it to extreme situations where normal transformers are infeasible, and there seems to be more focus on inference speed but not training/inference memory.
4. Related to Con 3, it might be nice to separate out training and inference (memory/speed).

Question for the authors:
for speed and peak memory, do you only measure at inference time? Is there any evaluation of training statistics like memory and time to convergence?

Typo:
This is the most comprehensive and extensives -> extensive

Overall, I think this would become an important bechmark for comparing efficient transformer approaches, and I would recommend its acceptance if the issues I mentioned above can be mitigated.

---

> ### Author Response · Authors · 2020-11-18
> **Response to Review**
>
> Thanks for the insightful comments and feedback, along with taking the time to review our paper! We are greatly appreciative of the extremely high quality feedback.
>
> Regarding the point on autoregressive decoding, this is a great suggestion. During the initial development of the benchmark, we deliberated a lot about including autoregressive decoding tasks in LRA. However, models such as Linformer or other models that use global memory tokens cannot be used in an autoregressive decoding setup. This makes it difficult to compare models fairly in overall evaluation.
>
> Ideally, as a starting point, we want a comparison with all models and so that no model is left out on any task. We fully agree that a transformer model that doesn’t support decoding is flawed. However, it is unclear of how we would then aggregate performance across different tasks in this setting.
>
> We can also add autoregressive decoding tasks (i.e., language modeling) results/comparison for models that are able to support autoregressive decoding. This since new autoregressive experiments may be quite extensive for a short rebuttal window and we do want to make a good effort to make sure these results are not rushed, we can commit to adding this by the camera ready deadline. Since our comment here on openreview is public, we would be held accountable for this commitment.
>
> Furthermore, when we do release our benchmark, we would also add generative modeling benchmarks (e.g., language modeling) for end users who are interested to try these models out.
>
> Regarding other modalities, this is indeed an interesting suggestion and we are very excited to try out different modalities such as audio, music, or speech!
>
> Regarding speed and memory, we currently report the training speed and memory usage. We will update the paper with inference speed. As for the training statistics, we find that most models converge with very similar curves. The only exception is Linformer, which often converges very quickly. Thanks for the suggestion! We will include details about convergence in the revised version.
>
> Once again, thanks for the great review and spending time to review our paper!
>
> Note: We will update the revised paper before 24th November, the deadline for revisions.

---

> > ### Comment · AnonReviewer4 · 2020-11-18
> > **Response to response**
> >
> > Thanks for the detailed response! It's good to know that the numbers refer to training time. It'd be great if you can update the paper with inference speed/memory, and include autoregressive decoding tasks.

---

### Author Response · Authors · 2020-11-24
**Summary of Updates in our Revision**

Dear Reviewers and Area Chair,

Thank you for the reviews, insightful comments and hard work!

We have modified our paper to take into consideration some of the feedback by reviewers. Specifically:

1. We added a section to the Appendix / Supplementary material that reports Inference/Eval speed. We also made the section on speed benchmarking more clear in the main paper and point readers to the appendix for the inference speeds. The inference speeds are not that much different (in terms of relative comparisons amongst models) when compared with the training speed.
2. We added a section that reports time to convergence of all xformer models in the supplementary material. This shows the time taken for each model to get up to N accuracy on the image benchmark.
3. We (slightly) expanded the section in related work on efficient transformers and provided more details.
4. We added more commentary in the experimental results discussion on how the tasks probes for a certain capability of the models and how different groups of model fare under this circumstance (e.g., on ListOps).
5. We added the random chance results to Table 1 to help understand how each model is doing/learning on each task.
6. We fixed several typos and made several cosmetic edits.

Once again, thanks for the reviews and insightful comments!

---

### Decision · Program_Chairs · 2021-01-07
**Final Decision**

**Decision:**

Accept (Poster)

**Comment:**

The paper attempts at providing a general benchmark for evaluating/analysis of long range transformer models, consisting of a 6 evaluation tasks. The main goal of the paper is to remove conflating factors such as pretraining from model performance and keeping the benchmark accessible. All reviewers agreed that these are important positive aspects of the paper and the presented analysis/results are useful.

While reviewers generally feel positive about the work, there are some critical concerns on how useful this benchmark is in practice, how generalizable are the results, and whether the benchmark is good at what is intended for. For example, the vanilla Transformer model performs very well on all the proposed tasks, making me question on what we can actually learn about long range dependencies through this benchmark. In addition, most tasks are synthetic and all models fail on 1 of the 6 proposed tasks.

Therefore, I think LRA should be viewed more as a tool for analysis, or as authors nicely put in their response, it should be viewed as a means to "encourage hypothesis driven research instead of hillclimbing or SOTA chasing.".
During discussion period with reviewers, while acknowledging the above-mentioned issues, this strength was highlighted as a valuable contribution. Therefore, given the general positive sentiment about the work, I'd recommend accept.